# Whole Blood Thromboelastometry by ROTEM and Thrombin Generation by Genesia According to the Genotype and Clinical Phenotype in Congenital Fibrinogen Disorders

**DOI:** 10.3390/ijms22052286

**Published:** 2021-02-25

**Authors:** Timea Szanto, Riitta Lassila, Marja Lemponen, Elina Lehtinen, Marguerite Neerman-Arbez, Alessandro Casini

**Affiliations:** 1Unit of Coagulation Disorders, Department of Hematology, Helsinki University Hospital Comprehensive Cancer Center, University of Helsinki, 00029 HUS Helsinki, Finland; riitta.lassila@kolumbus.fi (R.L.); anna-elina.lehtinen@hus.fi (E.L.); 2Research Program in Systems Oncology in Faculty of Medicine, University of Helsinki, 00029 HUS Helsinki, Finland; marja.lemponen@hus.fi; 3Department of Genetic Medicine and Development, Faculty of Medicine, University of Geneva, 1211 Geneva, Switzerland; Marguerite.Neerman-Arbez@unige.ch; 4Division of Angiology and Hemostasis, University Hospitals of Geneva, 1211 Geneva, Switzerland; Alessandro.Casini@hcuge.ch

**Keywords:** congenital fibrinogen defects, thrombin generation by Genesia, ROTEM

## Abstract

The outcome of congenital fibrinogen defects (CFD) is often unpredictable. Standard coagulation assays fail to predict the clinical phenotype. We aimed to assess the pheno- and genotypic associations of thrombin generation (TG) and ROTEM in CFD. We measured fibrinogen (Fg) activity and antigen, prothrombin fragments F1+2, and TG by ST Genesia^®^ with both Bleed- and ThromboScreen in 22 patients. ROTEM was available for 11 patients. All patients were genotyped for fibrinogen mutations. Ten patients were diagnosed with hypofibrinogenemia, nine with dysfibrinogenemia, and three with hypodysfibrinogenemia. Among the 17 mutations, eight were affecting the Fg γ chain, four the Fg Bβ chain, and five the Fg Aα chain. No statistical difference according to the clinical phenotypes was observed among *FGG* and *FGA* mutations. Median F1+2 and TG levels were normal among the different groups. Fg levels correlated negatively with F1+2 and peak height, and positively with lag time and time to peak. The pheno- and genotypes of the patients did not associate with TG. FIBTEM by ROTEM detected hypofibrinogenemia. Our study suggests an inverse link between low fibrinogen activity levels and enhanced TG, which could modify the structure–function relationship of fibrin to support hemostasis.

## 1. Introduction

Congenital fibrinogen disorders (CFD) are rare disorders affecting either the quantity (afibrinogenemia and hypofibrinogenemia) or the quality (dysfibrinogenemia) or both (hypodysfibrinogenemia) of fibrinogen (Fg) [1]. Hypofibrinogenemia and dysfibrinogenemia are the most frequent types of CFD, characterized by decreased levels of both Fg activity and antigen in the former, and discrepant levels of dysfunctional Fg in the latter. They usually result from heterozygous mutations affecting one of the three fibrinogen genes (i.e., *FGA*, *FGB*, or *FGG)* [2]. Hypodysfibrinogenemia is rarely reported. The diagnosis is based not only on a disproportional decrease of activity and antigen Fg levels, but also on specific molecular patterns [3].

The clinical course of CFD is often unpredictable. The clinical manifestations are very heterogeneous, from absence of symptoms to major bleeding or thrombosis (either arterial or venous) and pregnancy-related complications [4,5]. Moreover, asymptomatic individuals at the time of diagnosis may carry a lifetime risk of adverse outcomes. Although the diagnosis of most CFD is usually quite straightforward based on standard coagulation assays [6,7], these conventional tests fail to predict the clinical phenotype. Especially, the dysfunctional/abnormal Fg structure in dysfibrinogenemia is poorly assessed by standard coagulation assays. According to the recommendations of the Factor XIII and Fibrinogen Subcommittee of the Scientific Standardization Committee of the ISTH, the classification of CFD should be based on both the clinical phenotype and fibrinogen levels. In addition, specialized research laboratories should complete the fibrinogen work-up, providing insights into the properties of the fibrinogen variants [8].

During the last few years, a growing interest has evolved in the use of global hemostasis assays to measure the dynamics of the entire clotting and fibrinolysis process and to study the risks of bleeding and thrombosis under various clinical settings [9,10]. Until now, few attempts have been made to characterize the usefulness of thromboelastography in patients with CFD [11,12,13]. The influence of fibrinogen level on thrombin generation (TG) has also been investigated [14]. However, data in the setting of CFD—notably qualitative fibrinogen disorders—have limited to very few case reports [15].

Lately, a new TG analyzer (ST Genesia^®^, Stago) has been released as the first fully automated TG method for clinical routine laboratories. ST Genesia^®^ offers a set of reagents balanced for sensitivity to procoagulant and anticoagulant protein deficiencies [16].

We aimed to assess the clinical outcome by studying TG by ST Genesia^®^ and ROTEM in relation to various phenotypes, and to explore whether the clinical phenotypes are associated with genetic mutations.

## 2. Results

A total of 22 patients, 12 from Finland and 10 from Switzerland, were included. The median age of patients was 43.5 years (range 21–83 years). Most of the patients were female (77.3%). Fourteen patients were unrelated.

The clinical features, genotypes and Fg levels of the patients are presented in Table 1.

Nine patients were diagnosed with hypofibrinogenemia based on concordant Fg activity and antigen levels (ratios > 0.7), with median Fg activity level of 0.8 g/L (range 0.45–1.0 g/L) and median Fg antigen level of 1.0 g/L (range 0.40–1.2 g/L). Ten patients were diagnosed with dysfibrinogenemia based on discordant Fg activity and antigen levels (ratios <0.7), with median Fg activity level of 0.9 g/L (range 0.40–1.40 g/L) and median Fg antigen level of 2.40 g/L (range 1.80–3.10 g/L). In three patients, a definitive diagnosis of hypodysfibrinogenemia was confirmed upon genetic analysis. Their median Fg activity level was 0.8 g/L (range 0.70–1.08 g/L) and median Fg antigen level was 1.60 g/L (range 1.40–1.80 g/L). Bleeding tendency was classified with a bleeding score >3, which occurred in 5/22 (22.7%) patients, including three hypofibrinogenemia and two dysfibrinogenemia patients. Bleeding symptoms included menorrhagia (*n* = 5), vaginal bleeding during pregnancy (*n* = 1), bleeding after injury (*n* = 1), hemarthrosis (*n* = 1), postsurgical hemorrhage (*n* = 3). Two patients had only thrombotic complications (9%). One female with hypofibrinogenemia (HEL 1.3), obesity, and type 2 diabetes experienced a provoked deep venous thrombosis (DVT) and was treated with warfarin for three months. A patient with dysfibrinogenemia (HEL 2.1), obesity and dyslipidemia developed unprovoked pulmonary embolisms (PE) and DVT of the upper limb and was commenced on lifelong anticoagulant therapy with warfarin. Two patients had both bleeding and thrombotic complications. One female with hypodysfibrinogenemia (GEN 7) suffered from provoked PEs following a hysterectomy due to menorrhagia. She was treated by three months of low molecular weight heparin (LMWH). Another female with hypofibrinogenemia (HEL 3.1) and obesity experienced DVT of the right lower limb during intake of oral contraceptive at the age of 18, and later at the age of 36 two unprovoked DVTs, one of the left lower extremity, and three months later left iliofemoral DVT despite warfarin therapy. For the latest, she underwent iliac vein stenting under fibrinogen supplementation (Riastap, CSL Behring) and received a long-term antithrombotic treatment with LMWH and aspirin. She also suffered from menorrhagia but gave birth three times without major bleeding complications. She received LMWH prophylaxis throughout her pregnancies and pospartum up to 6 weeks. Thirteen patients were asymptomatic, of which 4/22 (18.2%) had hypofibrinogenemia, 7/22 (31.8%) dysfibrinogenemia, and 2/22 (9%) hypodysfibrinogenemia (Table 1).

Genetic analyses identified 14 missense mutations in 19 patients from 14 unrelated families, and three deletions in the remaining three patients. As shown in Table 1, almost all Finnish patients had missense mutations clustered in exons 2, 8, or 9 of *FGG*. The same double heterozygous *FGG* mutations, most likely carried on the same chromosome, p.Thr303Pro and p.Asp327His were discovered in two unrelated families (families 1 and 3, *n* = 5). In family 2, the *FGG* p.Tyr306Cys mutation was identified in both heterozygous and homozygous form. Interestingly, the patient HEL 2.2 with the homozygote form of the mutation was asymptomatic, while the patient 2.1 with the heterozygote form had thrombotic complications. More heterogeneous distribution was observed in the Swiss patients. Four had mutations affecting the Fg Aα chain, four the Fg Bβ chain and three the Fg γ chain. Of note, the asymptomatic patient GEN 6 had a mutation in *FGG* exon 8 p.Arg375Trp which is the causative mutation of fibrinogen Aguadilla, associated with a fibrinogen storage disease [17]. No statistical difference according to the clinical phenotypes was observed among *FGG* and *FGA* mutations.

### 2.1. Thrombin Generation

TG variables by ST Genesia^®^ (STG-BLS and STG-TS) are reported as normalized (Table 2) and absolute values (Figure 1, Figure 2 and Figure 3). Patients 2.1 and 3.1 with long-term anticoagulation and antiplatelet therapy were excluded from this study. None of the patients received oral contraception or hormonal therapy during the TG studies.

All median TG levels were within the normal reference range [16]. The inter-individual variation was up to 6-fold for peak height and up-to 5-fold for ETP among patients with STG-TS in the presence of thrombomodulin (TM). Variations were dependent on the clinical phenotype or Fg levels. The largest CV was observed for the peak height among asymptomatic patients and dysfibrinogenemia (6-fold and 5-fold as compared to 3-fold variation in normal donors).

In STG-BLS, lag time correlated positively with Fg activity (r = 0.6, *p* = 0.03), the peak height and ETP negatively with fibrinogen activity (r = −0.6, *p* = 0.03 and r = −0.7, *p* < 0.0001, respectively; Figure 4). In STG-TS with TM, there was a positive correlation between lag time, time to peak and Fg activity (r = 0.6, *p* < 0.002 and r = 0.5, *p* < 0.01) and a negative correlation between both ETP and Fg activity (r = −0.5, *p* = 0.03, and r = −0.5 and *p* = 0.02). However, the subgroup analysis according to the type of CFD and the clinical phenotype did not reveal any difference between the groups nor statistical difference between TG variables, clinical phenotype, type of CFD, and presence of TM (Figure 1, Figure 2 and Figure 3; data shown as absolute values).

There was no correlation among all the variables tested when comparing dysfibrinogenemic mutations affecting the thrombin binding site at the N-terminal end of the fibrinogen Aα chain (*FGA* exon 2) to mutations affecting the end-to-end alignment of fibrinogen or fibrin molecules in assembling polymers (*FGG* exon 8).

### 2.2. In Vivo Thrombin Generation

F1+2 fragment levels varied between 82 and 552 pM (reference 69–229 pM), indicating normal to somewhat increased in vivo TG (data not shown). The inter-individual variation was up to 7-fold among the patients. We demonstrated a negative correlation between F1+2 and both Fg antigen (r = −0.3, *p* < 0.006) and activity (r = −0.8, *p* = < 0.0001) levels (Figure 5). We further demonstrated a negative correlation between F1+2 and lag time by both STG-BLS (r = −0.3, *p* < 0.0003) and STG-TS in the absence of TM (r = −0.24, *p* < 0.001). The subgroup analysis according to CFD and clinical phenotype did not reveal any difference between the groups (Figure 6).

### 2.3. ROTEM

ROTEM analysis was carried out in 11 patients (7 hypo- and 5 dysfibrinogenemia, all from Finland) using the standard reagents. In hypofibrinogenemia, CT and MCF were markedly abnormal (Figure 7A). Median CT in hypofibrinogenemia compared with dysfibrinogenemia patient was 233 s (range 81–5414 s) vs. 55 s (range 50–132) (*p* = 0.0303). Median CT in hypofibrinogenemia was 76% prolonged as compared with dysfibrinogenemia. Median MCF in hypofibrinogenemia compared with dysfibrinogenemia patient was 4 mm (range 0–6 mm) vs. 13 mm (range 11–24 mm) (*p* = 0.0022), corresponding to 69%, lowered value in dysfibrinogenemia. MCF by FIBTEM correlated with Fg antigen (r = 0.6, *p* = 0.006) but not with activity (r = −0.2, *p* = 0.4). Moreover, there was a statistical difference among the patient’s clinical phenotype for the FIBTEM CT (*p* = 0.0003).

Median CT and MCF were all normal by EXTEM, INTEM, and APTEM. Median MCF in hypofibrinogenemia compared with dysfibrinogenemia patients was significantly reduced in all other ROTEM tests as follows: EXTEM hypofibrinogenemia 49 mm (below normal range) vs dysfibrinogenemia 61 mm (*p* = 0.003; normal range 58–63 mm), INTEM hypofibrinogenemia 48 mm (below normal) vs. dysfibrinogenemia 60 mm (*p* = 0.004; normal range 59–64 mm) and APTEM hypofibrinogenemia 52 mm vs. dysfibrinogenemia 62 mm (*p* = 0.007; normal range 50–72 mm (Figure 7B). We observed a strong uniform negative correlation between Fg activity and in overall MCFs by EXTEM (r = −0.6, *p* = 0.004), INTEM (r = −0.6, *p* = 0.004) and APTEM (r = −0.7, *p* = 0.002). A weaker negative correlation between Fg antigen and MCF by EXTEM (r = −0.5, *p* = 0.01), INTEM (r = −0.5, *p* = 0.01), and APTEM (r = −0.6, *p* = 0.009) was also found.

## 3. Discussion

In this study, we investigate for the first time the TG by ST Genesia^®^ in CFD and correlated biological and clinical phenotypes with ROTEM parameters. We observed a significant correlation between Fg levels and TG, both with ST Genesia^®^ in vitro and prothrombin fragments F1+2 reflecting FXa in vivo. An important association between the laboratory phenotype of CFD and FIBTEM was picked up.

The final fibrin clot has a highly heterogeneous structure, which is determined by both genetic and environmental factors [18]. Lower Fg concentrations increase bleeding risks due to formation of thicker fibrin fibers which are more susceptible to fibrinolysis. In contrast, abnormal structure of fibrin paradoxically increases thrombotic risk by impairing fibrin degradation by plasmin or facilitating clot fragmentation [18,19] and even enhancing TG due to poorly polymerizing fibrin [20]. Therefore, TG and other coagulation activation markers, such as prothrombin activation fragments, fibrin structure and consequences of the fibrinolytic system are of particular interest in CFD [21].

To our knowledge, this is the largest study reporting data on thrombin generation in CFD and the first study to assess the value of the ST Genesia^®^ device in the context of CFD. TG seems to be affected by the abnormal fibrin structures as Fg activity levels correlated with most TG variables. One of the most important questions is whether ST Genesia^®^ can help to identify the dysfibrinogenemia patients at the highest risk of thrombotic complications. Indeed, it could be that patients with the most decreased Fg activity have significantly higher ETP and peak height, and shortened lag time when challenged to low concentration of TF. Furthermore, in vivo TG was enhanced in patients with lower Fg levels, as occurs in a rare metabolic genetic disorder of lysinuric protein intolerance, LPI [21]. However, the clinical phenotype failed to be associated with TG variables. A more global approach taking also into account the fibrin clot structure including the capacity to make cross links may help to determine the heterogeneity of the clinical phenotype.

Only a few clinical studies explored non-conventional assays, such as thromboelastography (TEG) or ROTEM in predicting the clinical course of CFD. In these studies, TEG variables were affected by the causative mutations, but TEG failed to predict adverse events in most of the patients [11]. In one study, abnormal TEG results in non-pregnant women associated with an increased risk of obstetric complication during pregnancy [12]. One study on ROTEM showed that patients with dysfibrinogenemia have much higher median values of MCF than patients with hypodysfibrinogenemia [22], which may possibly reflect differences in the natural history of these disorders. This is in line with our observation as we found markedly abnormal median CT and CFT readings by FIBTEM in hypofibrinogenemia as compared to dysfibrinogenemia. Moreover, MCFs were significantly lower in the hypofibrinogenemia group than in dysfibrinogenemia according to all routine ROTEM tests used and correlated with Fg levels in all. Among them, FIBTEM MCF is of particular value to measure the contribution of Fg to the clot firmness, since cytochalasin D is first used to inactivate the platelets in the sample [23,24]. Here, on the other hand, the role of platelets remains obscure. Lower MCF readings in the patients such may reflect an impaired fibrin polymerization capacity. So far, no correlation between thromboelastometry and clinical phenotypes was demonstrated. Instead, in our subgroup analysis, we observed an association between the clinical phenotype of CFD and FIBTEM. Larger prospective studies are warranted to further validate and explore the mechanisms underlying these findings.

The spectrum of molecular abnormalities in CFD is broad, resulting in several subtypes of fibrinogen disorders with specific biological and clinical features [17]. Only a few fibrinogen variants are definitely associated with a clinical course. Thrombosis-related dysfibrinogenemia leads to a strong thrombotic phenotype, which usually is present in young adult with a thrombotic familial history [25]. A few fibrinogen mutations, clustered in exons 8 and 9 of *FGG*, cause fibrinogen deposit in endoplasmic reticulum, which in turn can give rise to liver failure [26]. However, in dysfibrinogenemia genotype cannot predict the bleeding risk. As confirmed by our observations, the individual range of TG variables was high without any correlation according to the site of fibrinogen mutation (*FGA*; Aα chain vs. γ chain). It is likely that besides fibrinogen variants, other relatively common genetic polymorphisms or epigenetics in coagulation and fibrinolytic pathways may affect the fibrin clot structure and therefore act as modifiers of the blood clot function in dysfibrinogenemia.

Our study is limited by the patient numbers, being too small to allow any further conclusions on the clinical phenotype vs. thrombin generation. Nevertheless, it presents a cohort of 22 CFD patients, as higher numbers are difficult to reach considering the relatively low prevalence of the disease and pre-analytical requirements for the TG assay.

In conclusion, our data suggests that ROTEM has a high sensitivity towards detection of hypofibrinogenemia. While the assessment of ROTEM, MCF, and especially FIBTEM may help to discriminate patients with hypo- or dysfibrinogenemia, its effectiveness in predicting the clinical phenotypes has to be confirmed on larger groups of patients. Our study suggests an inverse link between low fibrinogen activity levels and enhanced TG, likely including poor fibrin build-up which could modify the structure–function relationship of fibrin to support hemostasis. These similarities have been observed in other clinical inherited conditions such as FXIII deficiency and LPI with enhanced TG but overly expressed fibrinolysis.

## 4. Materials and Methods

Our study was conducted in accordance with the Declaration of Helsinki and approved by the relevant local institutional ethical committees (191/13/03/01/2014; 24 September 2014). All patients consented to take part in the study. The study included patients and their relatives with confirmed diagnosis of CFD from Helsinki, Finland (Coagulations Disorders Unit, Comprehensive Cancer Center, Helsinki University Hospital) and from Geneva, Switzerland (Division of Angiology and Hemostasis, University Hospitals of Geneva) between 2018 and 2020. CFD was classified based on the activity and antigen levels of Fg and genotype according to the International Society of Thrombosis and Hemostasis (ISTH) classification [8]. Symptoms of bleeding and/or thrombosis were recorded from the medical files. The bleeding score was evaluated by the ISTH bleeding assessment tool [27].

### 4.1. Blood Sampling

Fasting peripheral venous blood samples were collected under stable conditions at a time remote from acute infections, bleeds or surgeries, between 8–10 am into vacutainer tubes containing: (1) EDTA to measure blood cell counts and hemoglobin levels, and (2) sodium citrate (109 mM sodium citrate, 3.2%, BD-Plymouth) for routine coagulation, TG and ROTEM.

### 4.2. Coagulation Studies

Citrated blood samples were centrifuged to isolate plasma and frozen in aliquots at −70 °C, if not tested immediately. Plasma levels of functional Fg were measured by the Clauss method using the HemosIL Fibrinogen activity (IL QFA Thrombin) on an ACL Top Analyzer or Multifibren U (Siemens, Marburg, Germany) on a BCS^®^ XP coagulometer. Levels of total Fg antigen were measured by a latex immunoassay (Liaphen Fibrinogen, Hyphen BioMed, Neuville sur Oise, France) on a BCS^®^ XP coagulometer (Siemens, Germany).

Prothrombin time (PT; Owren method, Axis-Shield, Oslo, Norway), activated partial thromboplastin time (APTT; Actin FSL reagent) and thrombin time (Hemoclot Thrombin Time, Hyphen BioMed, France) were assessed in citrated plasma on an ACL Top Analyzer analyzer. Prothrombin fragments (F1+2) were measured by an enzyme immunoassay (Enzygnost^®^ F1+2, monoclonal, Siemens Healthcare Diagnostics, Marburg, Germany.

### 4.3. DNA Preparation and Genetic Analysis

Genomic DNA was extracted from fresh blood samples using standard procedures. Fibrinogen mutation screening was performed by PCR amplification of fibrinogen coding sequences and intron-exon junctions followed by Sanger sequencing (samples until 2019) or by whole exome sequencing performed at the Health 2030 Genome Centre at Campus Biotech, Geneva using Twist Core +Refseq Exome reagents. Variant calling was filtered for a panel of genes of the coagulation and fibrinolytic pathways, including the fibrinogen genes. Variant analysis was performed using the Ensembl Variant Effector Predictor (VEP) tool: http://www.ensembl.org/Homo_sapiens/Tools/VEP (accessed on 1 December 2020). Mutations were confirmed by polymerase chain reaction (PCR) amplification of by sequencing as previously described [28]. According to guidelines of the Human Genome Variation society, mutations are reported with nucleotides numbered from the A in the ATG (Met) initiator codon and amino acids numbered from Met 1 [29].

### 4.4. Thrombin Generation

TG was measured in platelet-poor (PPP) plasma by using the novel, fully-automated ST Genesia^®^ Analyzer (Diagnostica Stago S.A.S., Asnières-sur-Seine, France), which is based on the reference Calibrated Automated Thrombogram method [14]. PPP was prepared by double centrifugation at 2500× *g* for 10 min at room temperature within 2 h of blood collection [14]. The assessment is based on measurement of fluorophore aminomethylcoumarin (AMC) generation, after adding a standard amount of human recombinant tissue factor and synthetic phospholipids to induce TG in the test plasma. AMC generation is monitored every 15–20 s at 450 nm. All measurements were carried in duplicate by using two types of reagents: the STG^®^-BleedScreen (STG-BLS, Stago, Asnières-sur-Seine, France and the STG^®^-ThromboScreen (STG-TS, Stago, Asnières-sur-Seine, France). TG with the STG-BLS assay was triggered by a mixture of procoagulant phospholipids and low picomolar level of human tissue factor (TF), balanced for sensitivity to procoagulant factor deficiencies while minimizing contact activation. TG with the STG-TS assay was initiated by a mixture of procoagulant phospholipids and medium picomolar level of human TF in the presence or absence of TM, balanced for sensitivity to deficiencies in natural anticoagulants, without interfering contact activation. Since the addition of corn trypsin inhibitor upon blood collection did not make a difference in the data we are reporting the results obtained only in the citrated samples.

We followed endogenous thrombin potential (ETP), peak thrombin (peak height), lag time, and time to peak. Both absolute and normalized values of each TG parameter is provided. Normalization of each TG parameter is based on a reference plasma for each test aiming to reduce the interlaboratory variability as well as the variability between different measurement runs. We used standard human plasma in addition to the reference plasma TG levels provided by the manufacturer to assess TG in normal plasma.

### 4.5. Rotational Thromboelastometry

Thromboelastometry in whole blood was measured within 2 h of blood collection by ROTEM [30]. The standard INTEM, EXTEM, FIBTEM, and APTEM activation methods were performed by automated pipette program. We analyzed clotting time (CT, s) and maximum clot firmness (MCF, mm). The reference ranges were: FIBTEM CT 46–84 s and MCF 6–21 mm, INTEM CT 161–204 s and MCF 51–69 mm; EXTEM CT 50–80 s, and MCF 55–72 mm; APTEM CT 41–80 s, and MCF 52–71 mm.

### 4.6. Prothrombin Fragments F1+2

Circulating prothrombin fragments (F1+2) were measured in plasma with Enzygnost^®^ ELISA (Siemens). The manufacturer’s reference ranges (5th to 95th percentiles) are 69–229 pmol/L.

### 4.7. Statistical Analysis

The recorded data are expressed as median and range. The significance of difference between two distinct data groups was examined with Mann–Whitney tests. The significance of difference between more groups was estimated by one-way analysis of variance (ANOVA). The correlation between two variables was measured using Spearman r test. Calculations were performed with Prism 7.0 or 9.0 (GraphPad Software, La Jolla, CA, USA).

## Figures and Tables

**Figure 1 ijms-22-02286-f001:**
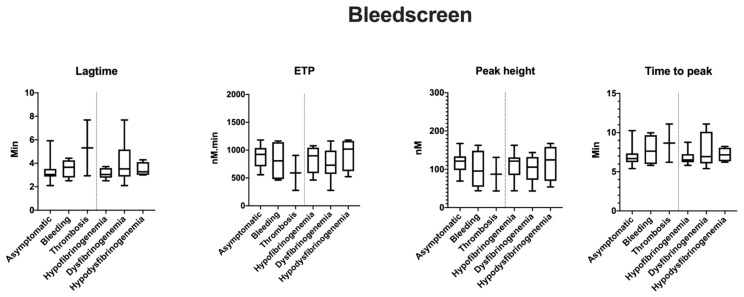
Mean TG variables with STG-BLS (bleedscreen) according to clinical and laboratory phenotypes. TG values are shown for 19 patients as follows: asymptomatic *n* = 13; bleeding *n* = 4, thrombosis *n* = 2; hypofibrinogenemia *n* = 8; dysfibrinogenemia *n* = 8 and hypodysfibrinogenemia *n* = 3.

**Figure 2 ijms-22-02286-f002:**
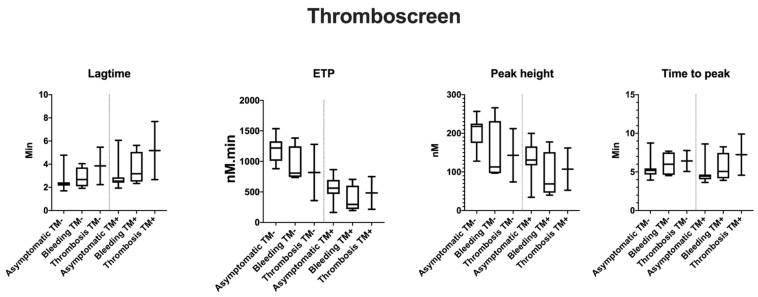
Mean TG variables with STG-TS (thromboscreen) according to clinical phenotype and the supplementation of TM. TG values are shown for 19 patients as follows: asymptomatic *n* = 13; bleeding *n* = 4, thrombosis *n* = 2; hypofibrinogenemia *n* = 8; dysfibrinogenemia *n* = 8; and hypodysfibrinogenemia *n* = 3.

**Figure 3 ijms-22-02286-f003:**
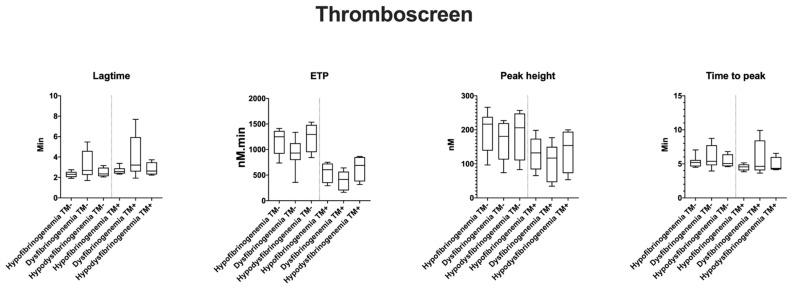
Mean TG variables with STG-TS (thromboscreen) according to laboratory phenotype and the supplementation of TM. TG values are shown for 19 patients as follows: asymptomatic *n* = 13; bleeding *n* = 4, thrombosis *n* = 2; hypofibrinogenemia *n* = 8; dysfibrinogenemia *n* = 8; and hypodysfibrinogenemia *n* = 3.

**Figure 4 ijms-22-02286-f004:**
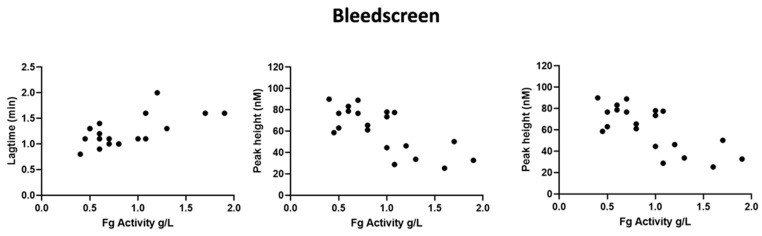
Correlation of TG variables with STG-BLS (bleedscreen) with fibrinogen activity. The data are shown TG values are shown for 19 patients as follows: asymptomatic *n* = 13; bleeding *n* = 4, thrombosis *n* = 2; hypofibrinogenemia *n* = 8; dysfibrinogenemia *n* = 8; and hypodysfibrinogenemia *n* = 3.

**Figure 5 ijms-22-02286-f005:**
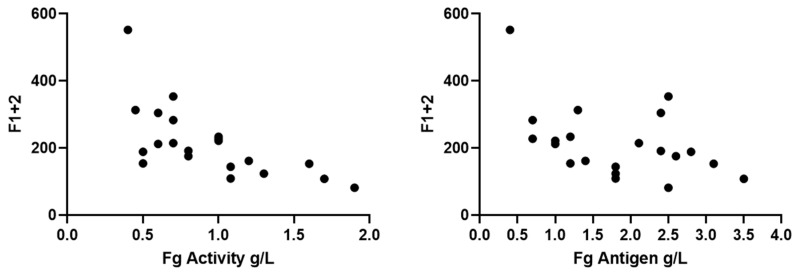
Correlation of F1+2 levels with Fibrinogen activity and antigen. Data are shown for 20 patients as follows: asymptomatic *n* = 13; bleeding *n* = 5, thrombosis *n* = 2; hypofibrinogenemia *n* = 9; dysfibrinogenemia *n* = 8 and hypodysfibrinogenemia *n* = 3.

**Figure 6 ijms-22-02286-f006:**
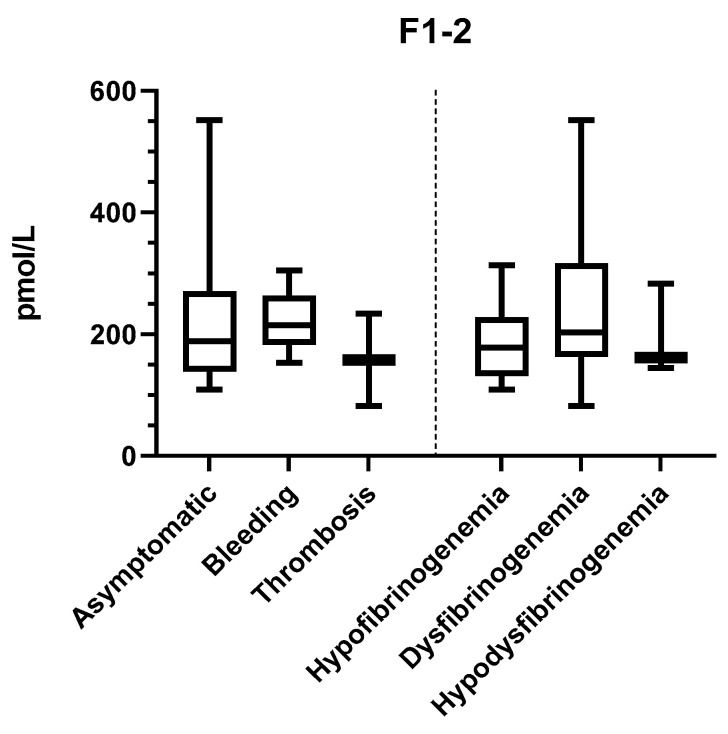
Mean F1+2 levels according to the clinical and laboratory phenotypes. F1+2 values are shown for 20 patients as follows: asymptomatic *n* = 13; bleeding *n* = 5, thrombosis *n* = 2; hypofibrinogenemia *n* = 9; dysfibrinogenemia *n* = 8; and hypodysfibrinogenemia *n* = 3.

**Figure 7 ijms-22-02286-f007:**
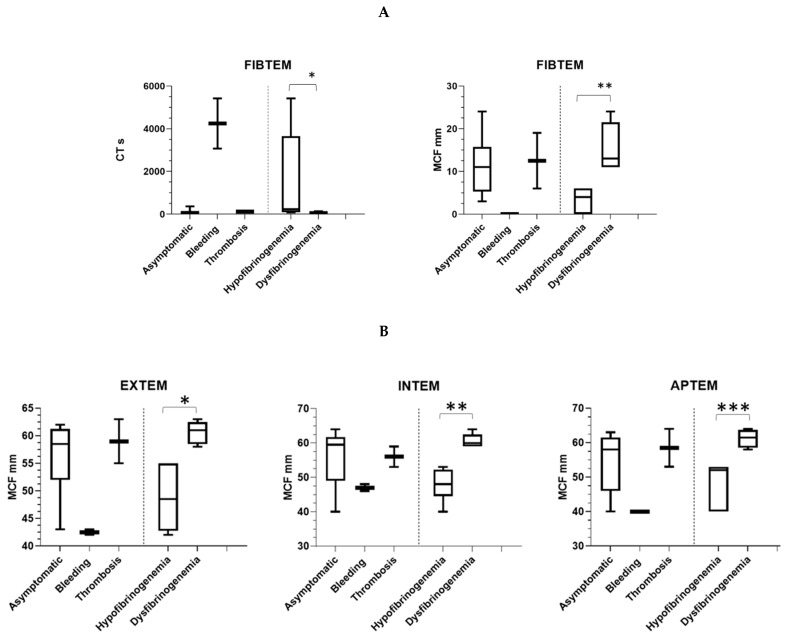
ROTEM profiles according to clinical and laboratory phenotypes. (**A**). FIBTEM CT and MCF are shown for 110 patients as follows: asymptomatic *n* = 7; bleeding *n* = 2, thrombosis *n* = 2; dysfibrinogenemia *n* = 6 and dysfibrinogenemia *n* = 5. * CT was markedly prolonged in hypo- in comparison with dysfibrinogenemia (*p* = 0.0303). ** Fibrin polymerization capacity (clot firmness) in FIBTEM was decreased (MCF ≤ 6 mm) in hypofibrinogenemia as compared with dysfibrinogenemia (*p* = 0.0022). (**B**). EXTEM, INTEM, and APTEM MCF are shown for 11 patients as follows: asymptomatic *n* = 7; bleeding *n* = 2, thrombosis *n* = 2; dysfibrinogenemia *n* = 6; and dysfibrinogenemia *n* = 5 MCF was markedly decreased in hypo- in comparison with dysfibrinogenemia in EXTEM (* *p* = 0.003), INTEM (** *p* = 0.004), and APTEM (*** *p* = 0.007).

**Table 1 ijms-22-02286-t001:** Clinical manifestations, genotypes, and phenotypic characteristics according to traditional fibrinogen assays.

	Patient	Age	Gender	Phenotypes	Fg Act	Fg Ag	Type	Gene	Mutation	Family Members
Family 1	HEL 1.1	72	Female	Asymptomatic	1.3	1.3	Hypo ^1^	*FGG*, exon 8	het Thr303Pro and het Asp327His	Thrombosis
	HEL 1.2	24	Male	Asymptomatic	0.7	0.7	Hypo ^1^	*FGG*, exon 8	het Thr303Pro and het Asp327His	
	HEL 1.3	48	Female	Thrombosis	1.0	1.2	Hypo ^1^	*FGG*, exon 8	het Thr303Pro and het Asp327His	
Family 2	HEL 2.1	82	Male	Thrombosis	1.2	2.5	Dys ^2^	*FGG*, exon 8	het Tyr306Cys	Thrombosis
	HEL 2.2	58	Male	Asymptomatic	1.6	2.5	Dys ^2^	*FGG*, exon 8	homozygous Tyr306Cys	
	HEL 2.3	83	Male	Asymptomatic	1.7	3.5	Dys ^2^	*FGG*, exon 8	het Tyr306Cys	
Family 3	HEL 3.1	40	Female	Bleeding and thrombosis	1.0	1.7	Hypo ^1^	*FGG*, exon 8	het Thr303Pro and het Asp327His	Bleeding and thrombosis *
	HEL 3.2	30	Female	Bleeding	<1	0.7	Hypo ^1^	*FGG*, exon 8	het Thr303Pro and het Asp327His	
Not related	HEL 4	35	Female	Asymptomatic	0.8	1.8	Hypodys ^3^	*FGG,* exon 9	het Trp395Leu	Thrombosis
	HEL 5	38	Female	Asymptomatic	<1	2.6	Dys ^2^	*FGG,* exon 9	het c.1283-1284 del TG	Thrombosis
	HEL 6	43	Female	Asymptomatic	1.9	2.4	Dys ^2^	*FGA,* exon 2	het Leu28Pro	NA
	HEL 7	47	Female	Bleeding	<1	1.0	Hypo ^1^	*FGG,* exon 9	het Thr397Ile	Bleeding
	GEN 1	37	Female	Bleeding	0.6	2.4	Dys ^2^	*FGG*	Arg301His	Bleeding
	GEN 2	39	Female	Asymptomatic	0.5	1.8	Dys ^2^	*FGB*	c.402_410 del GGAAGCTGT	NA
	GEN 3	49	Female	Bleeding	0.7	2.1	Dys ^2^	*FGA*	Arg35His	Bleeding
	GEN 4	30	Male	Asymptomatic	0.7	1.4	Hypo ^1^	*FGB*	Thr407Met	Bleeding
	GEN 5	51	Male	Asymptomatic	0.5	2.8	Dys ^2^	*FGA*	Arg38Gly	Asymptomatic
	GEN 6	44	Male	Asymptomatic	1.1	1.2	Hypo ^1^	*FGG*	Arg375Trp	Asymptomatic
	GEN 7	57	Female	Bleeding and thrombosis	1.1	1.8	Hypodys ^3^	*FGB*	Arg285Leu	Bleeding
	GEN 8	31	Male	Asymptomatic	0.4	0.4	Hypo ^1^	*FGA*	11 kb del	Asymptomatic
	GEN 9	53	Female	Asymptomatic	0.6	3.1	Dys ^2^	*FGA*	Arg35His	NA
	GEN 10	21	Female	Bleeding	0.9	1.0	Hypo ^1^	*FGB*	Arg294Ser	Asymptomatic

^1^ Hypofibrinogenemia, ^2^ Dysfibrinogenemia, ^3^ Hypodysfibrinogenemia.

**Table 2 ijms-22-02286-t002:** Mean normalized TG variables with STG-BLS and STG-TS in the absence of thrombomodulin according to clinical and laboratory phenotypes.

**STG-BLS Normalized**	**Lag Time, Ratio (Range)**	**ETP, % (Range)**	**Peak Height, % (Range)**	**Time to Peak, Ratio (Range)**
Hypofibrinogenemia (*n* = 8)	1.2 (0.9–1.4)	82.7 (41.6–101.5)	88,5 (30.6–113.2)	1.1 (1.0–1.4)
Dysfibrinogenemia (*n* = 8)	1.1 (0.8–2.0)	80.4 (49.2–111.2)	81.4 (46.6–104.5)	1.1 (0.9–1.7)
Hypodys-fibrinogenemia (*n* = 3)	1.0 (1.0–1.6)	83.0 (49.2–105.0)	81.4 (39.2–121.7)	1.1 (1.0–1.3)
*Reference range **	*0.9*–*1.3*	*57*–*114*	*51*–*99*	*0.9*–*1.3*
**STG-TS-TM Normalized**	**Lag Time, Ratio (Range)**	**ETP, % (Range)**	**Peak Height, % (Range)**	**Time to Peak, Ratio (Range)**
Hypofibrinogenemia (*n* = 8)	1.1 (0.9–1.3)	79.0 (46.3–90.5)	75.1 (32.7–90.0)	1.2 (1.1–1.7)
Dysfibrinogenemia (*n* = 8)	1.1 (0.8–2.2)	65.0 (48.1–96.5)	62.2 (33.7–78.8)	1.2 (0.9–2.0)
Hypodys-fibrinogenemia (*n* = 3)	1.1 (1.1–1.5)	80.8 (52.9–83.1)	65.5 (28.8–89.1)	1.2 (1.1–1.6)
*Reference range **	*1.0*–*1.3*	*59*–*99*	*45*–*102*	*1.0*–*1.5*

* Reference range as according to the literature [16].

## Data Availability

The data presented in this study are available on request from the corresponding author.

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
