# Peer review of "Whole Blood Thromboelastometry by ROTEM and Thrombin Generation by Genesia According to the Genotype and Clinical Phenotype in Congenital Fibrinogen Disorders"

_ijms, 2021, doi:10.3390/ijms22052286_

Round 1

Reviewer 1 Report

Szanto et al. assessed the clinical outcome of 22 patients with congenital fibrinogen disorders by studying thrombin generation by ST Genesia and ROTEM analysis in relation to various phenotypes, and wanted to explore whether the clinical phenotypes are associated with genetic mutations.

The manuscript is well-written and appears to be straightforward. However, the novelty of the findings and how much this paper advances the field is unclear and has to be better stressed by the authors.

Major comments:

The state-of-the art in the field and what this paper add to the field need to be clearly formulated.

The conclusion of the paper written at the end of the abstract differs from the conclusion delivered at the end of the discussion: it needs to be harmonized.

At the time of thrombin generation measurement, were the patients taking any therapy that may affect this analysis (for example, anticoagulants, antiplatelet drugs, oral contraception, hormonal therapy)? This information has to be provided in the manuscript.

Table 2: From where comes the “reference range”?

The correlation studies displayed in the text (p. 5, lines 123-127; p. 7 lines: 148-150) are relevant results that might deserve to be presented in a figure.

p. 5, lines 130-131: The statement regarding patient 3.1 is unclear. Was the patient under anticoagulation? How the authors explain this result.

Minor comments:

p 4., line 106: there are two consecutive “the”. Please correct.

Table 2: decimal numbers comprise either “,” or “.”. I suggest to replace the “,” by “.”.

The discussion would gain being shorter and more focused.

Figure 1-5: In order to support clarity, the authors may indicate the reference ranges on the figures.

Author Response

Dear Dr. Makayla Li and Dear Reviewers,

We are sincerely grateful for your editorial comments and the reviewers’ remarks on our manuscript ‘Whole blood thromboelastometry by ROTEM and thrombin generation by Genesia according to the genotype and phenotype in congenital fibrinogen disorders’ by Szanto et al. We have accordingly made the suggested corrections. Please find the detailed response to each point raised by reviewers. You will find below our response marked with italics. Consequently, we provide tmodifications by using track change mode in the revised version of the manuscript. 

We hope that with these changes the current manuscript will meet the requirements for publication in the International Journal of Molecular Sciences.

On behalf of the co-authors,

Timea Szanto and Alessandro Casini

Reply to Reviewer: 1

We thank the reviewer for careful reading our manuscript and for the helpful comments and valuable suggestions which helps us in improving the manuscript.

Major comments:

1.The state-of-the art in the field and what this paper add to the field need to be clearly formulated.

We acknowledge this important issue. We have now added the recommendations of the Factor XIII and Fibrinogen Subcommittee of the Scientific Standardization Committee of the ISTH regarding diagnosis and classification of CFD, commented in the introduction part of the manuscript, page 2, lines 50-52: ‘’According to the recommendations of the Factor XIII and Fibrinogen Subcommittee of the Scientific Standardization Committee of the ISTH, the classification of CFD should be based on both the clinical phenotype and fibrinogen levels In addition, specialized research laboratories should complete the fibrinogen work-up, providing insights into the properties of the fibrinogen variants [8].’’ Data on thrombin generation, especially in dysfibrinogenemia, are very limited. We have added this sentence page 2, lines 56-59: “The influence of fibrinogen level on thrombin generation (TG) has also been investigated (HC Hemker Pathophysiol Haemost Thromb 2003;33:4–15). However data in the setting of CFD, notably qualitative fibrinogen disorders, have limited to very few case reports (S Bouvier Thromb Haemost 2018;118:2006–2008)”. Thus, it is likely that our study is the largest reporting data on CFD. Moreover, to our knowledge, this is the first study assessing the value of the ST Genesia (a relatively new thrombin generation assay) in a constitutional coagulopathy. 

2.The conclusion of the paper written at the end of the abstract differs from the conclusion delivered at the end of the discussion: it needs to be harmonized.

Thank you for pointing this out. We have now harmonized the conclusions of the abstract with the message delivered at the end of the discussion by making the following changes to the abstract, page 1, line 29: ‘’ FIBTEM by ROTEM detected hypofibrinogenemia and associated with the clinical phenotype.’’

3.At the time of thrombin generation measurement, were the patients taking any therapy that may affect this analysis (for example, anticoagulants, antiplatelet drugs, oral contraception, hormonal therapy)? This information has to be provided in the manuscript.

Thank you for this important question.

Two patients were on anticoagulant therapy at the time of thrombin generation studies: patient 2.1 on lifelong warfarin therapy and patient 3.1 on long-term low molecular weight therapy and aspirin. This information is given in the main text on page 4:

- lines 85-87 (patient 2.1): ‘’A patient with dysfibrinogenemia (HEL 2.1), obesity and dyslipidemia developed unprovoked pulmonary embolisms (PE) and DVT of the upper limb and was commenced on lifelong anticoagulant therapy with warfarin.’’

-lines 92-94 (patient 3.1): ‘’For the latest, she underwent iliac vein stenting under fibrinogen supplementation (Riastap, CSL Behring) and received a long-term antithrombotic treatment with LMWH and aspirin.’’

These patients are now excluded from the TG studies and accordingly changes were made to the revised manuscript as follows:

-page 5, lines 113-116: ‘’Patients 2.1 and 3.1 with long-term anticoagulation and antiplatelet therapy were excluded from this study. None of the patients received oral contraception or hormonal therapy during the TG studies.’’

-mean TG values presented in the table 1 were recalculated after excluding patients 2.1 and 3.1

- the correlations were re-calculated and accordingly changes were made to the main text on page 5: ‘’ In STG-BLS, lag time correlated positively with Fg activity (r=0.6, p=0.03), the peak height and ETP negatively with fibrinogen activity (r=-0.6, p=0.03 and r=-0.7, p <0.0001, respectively; Figure 1). In STG-TS with TM, there was a positive correlation between lag time, time to peak and Fg activity (r=0.6, p<0.002 and r=0.5, p<0.01) and a negative correlation between both ETP and Fg activity (r=-0.5, p=0.03 and r=-0.5 and p=0.02).’’

4.Table 2: From where comes the “reference range”?

The reference ranges is according to the literature as recommended by the manufacturer.

Calzavarini  S,  Brodard J Quarroz C  Maire L, Nützi R Jankovic J  Rotondo LC Giabbani E  Fiedler Michael Nagler GM Angelillo-Scherrer A.  Thrombin generation measurement using the ST Genesia Thrombin Generation System in a cohort of healthy adults: Normal values and variability. Res Pract Thromb Haemost 2019 Jul 18;3(4):758-768. doi: 10.1002/rth2.12238 

5.The correlation studies displayed in the text (p. 5, lines 123-127; p. 7 lines: 148-150) are relevant results that might deserve to be presented in a figure.

We agree. These data are now presented as figure 1 on page 6 and figure 4 on page 8.

6.p. 5, lines 130-131: The statement regarding patient 3.1 is unclear. Was the patient under anticoagulation? How the authors explain this result.

-Thank you for this relevant point. Patient 3.1 was on LMWH and Aspirin treatment, so we have now excluded here from the TG studies. However, the antithrombotic therapy does not necessary completely explain the lack of TG in the patient. Of note, we have a dysfibrinogenemia patient with severe thrombotic phenotype on lifelong LMWH and Aspirin therapy (not included in this study) with clearly measurable TG levels. We believe that the fibrinogen structure together with individual factors may affect TG in patients with qualitative CFD.

Minor comments:

p 4., line 106: there are two consecutive “the”. Please correct.

-We have now corrected and removed to extra ‘’the’’ from the text.

Table 2: decimal numbers comprise either “,” or “.”. I suggest to replace the “,” by “.”.

-We have replaced “,” by “.” both in tables 1 and 2.

The discussion would gain being shorter and more focused.

-Alessandro and Riitta: is there something we could remove from the discussion part?

We have now remowed the following statements from the discussion part, first paragraph:

So far, there is only limited data on TG in CFD, mainly on afibrinogenemia. An increase TG has been reported in afibrinogenaemic patients having a thrombotic phenotype [20]. After fibrinogen supplementation, the TG returned to control values, reflecting the thrombin binding to fibrin as one thrombin-clearance mechanism [20,21].

Figure 1-5: In order to support clarity, the authors may indicate the reference ranges on the figures.

The reference ranges for Genesia are provided as normalized values by the the manufactures. Our data as normalized values vs normalized reference range is presented in table 2. The figures represent the absolute values and the calculated means for each TG parameter. In these figures we compare the TG values between the different CFD patient groups according to their clinical or laboratory phenotype.

Reviewer 2 Report

Authors pulled up an interesting piece of work to evaluate the genotypic and phenotypic associations with CFD. The revolutionary innovations in the detection platforms were evaluated by the authors and correlations were made to the existing platforms. Further, the results supported the observation that there is an inverse relationship between low fibrinogen activity levels and enhanced TG. The article needs few minor revisions, 1) Did the authors do any power analysis before considering 22 subjects for this study? Were the diagnostic evaluations blinded? 2) In Table 1, please replace the header "relatives" with an appropriate word. This is a bit misleading. 3) Also in Table 1, for the first column use a proper header and merge the cells appropriately for each family to avoid confusion.

Author Response

Dear Dr. Makayla Li and Dear Reviewers,

We are sincerely grateful for your editorial comments and the reviewers’ remarks on our manuscript ‘Whole blood thromboelastometry by ROTEM and thrombin generation by Genesia according to the genotype and phenotype in congenital fibrinogen disorders’ by Szanto et al. We have accordingly made the suggested corrections. Please find the detailed response to each point raised by reviewers. You will find below our response marked with italics. Consequently, we provide tmodifications by using track change mode in the revised version of the manuscript. 

We hope that with these changes the current manuscript will meet the requirements for publication in the International Journal of Molecular Sciences.

On behalf of the co-authors,

Timea Szanto and Alessandro Casini

Reply to Reviewer 2

We thank the reviewer for careful reading our manuscript and for the helpful comments and valuable suggestions which helps us in improving the manuscript.

1) Did the authors do any power analysis before considering 22 subjects for this study? Were the diagnostic evaluations blinded?

Thank you for this important comment. Since very limited data is published on the TG in CFD no specific power calculation was possible. Of course, our study is limited by the patient numbers, being too small to allow any further analysis. However, our results could be used as reference for next larger studies. The diagnostic evaluations were blinded.

2) In Table 1, please replace the header "relatives" with an appropriate word. This is a bit misleading.

We have now replaced the word "relatives" by "Family members" in Table 1.

3) Also in Table 1, for the first column use a proper header and merge the cells appropriately for each family to avoid confusion.

We have now merged the cells appropriately for each family to avoid confusion.
